# Association Between Stride Parameters and Racetrack Curvature for Thoroughbred Chuckwagon Horses

**DOI:** 10.3390/s25237376

**Published:** 2025-12-04

**Authors:** Matthijs van den Broek, Zoe Y. S. Chan, Charlotte De Bruyne, Karelhia Garcia-Alamo, Sara Skotarek Loch, Thilo Pfau

**Affiliations:** 1Faculty of Kinesiology, University of Calgary, Calgary, AB T2N 1N4, Canada; 2Faculty of Mechanical Engineering, Delft University of Technology, 2628 CD Delft, The Netherlands; 3Faculty of Veterinary Medicine, University of Calgary, Calgary, AB T2N 1N4, Canada

**Keywords:** horse, gallop, chuckwagon, global navigation satellite system, speed, stride length, stride frequency, curve

## Abstract

Increased risk of musculoskeletal injury in galloping racehorses has been linked to decreased stride length and reduced speed over consecutive races prior to the injury. As racetrack curvature influences horses’ maximal speed, we hypothesized it also affects stride parameters. During training sessions, twenty-eight wagon-pulling Thoroughbred Chuckwagon horses were equipped with Global Navigation Satellite System (GNSS) loggers, allowing for identification of speed, stride length (SL) and stride frequency (SF), and average speed, SL and SF were calculated for consecutive 100 m sections. Effects of curvature on speed were investigated with a linear mixed model with speed as output variable, curvature as fixed factor, and horse as random factor. Effects of curvature and speed on stride parameters were investigated with linear mixed models with output variables SL and SF, continuous covariates speed, curvature, and the two-way interaction between curvature and speed as fixed factors, and horse as random factor. Curvature was associated with a significant increase in speed (*p* = 0.004), decrease in SL (*p* < 0.001) and increase in SF (*p* < 0.001), and for SL and SF the magnitude of these effects was dependent on speed (*p* < 0.001). At a curvature of 60° per 100 m, an increase in speed of 0.264 m/s was found compared to the straight, although this effect is likely confounded by fatigue. At the median speed of 14.5 m/s and a curvature of 60° per 100 m, a SF increase of 0.053 Hz (+2.4%) and a SL reduction of 0.137 m (−2.1%) was found compared to the straight. This is in the same order of magnitude as the 0.10 m SL reduction over consecutive races previously associated with increased injury risk. We conclude that, in Chuckwagon horses, interactions between speed and curvature are affecting stride parameters that have previously been identified as predictors of musculoskeletal injuries.

## 1. Introduction

Running along a curved trajectory is fundamentally different from running in a straight line due to the added necessity to generate centripetal force. Unlike greyhounds [1], both humans [2] and racehorses [3] have a lower maximum speed in the curve than on the straight, possibly due to differences in force generation between the inside and outside leg [2]. Also a shorter step length [4,5,6] in the curve compared to the straight has been observed in humans during sprint exercise on athletics tracks. In horses, Parkes et al. [7] reported that Thoroughbred horses display a longer stride duration during curve running compared to gallop on the straight. However, potential changes in speed were not taken into account in that study.

In Thoroughbred racehorses, retrospective investigations have shown an association between decreasing speed and stride length over consecutive races, and impending musculoskeletal injury [8]. The risk of musculoskeletal injury was reported to increase by 18% for each 0.1 m/s decrease in speed [8], and by 11% for each 0.10 m decrease in stride length [8], opening up the possibility to create stride parameter-based injury prediction models. However, to differentiate between changes in stride parameters due to impending injury or due to other, possibly confounding, factors, it is important to develop detailed quantitative models describing their association with stride parameters. Interestingly, a recent prospective study found a comparatively low accuracy for detecting horses at risk of musculoskeletal injury through stride parameter analysis [9]. This is likely related to the multi-factorial nature of stride parameter adaptations.

For example, between-race factors related to associations between racing conditions and stride parameters include track moisture and hardness [10], as well as surface type. Stride parameter adaptations to track hardness appear to be surface specific, with shorter gallop strides on hard compared to soft surfaces on dirt racetracks [10], while on turf racetracks the opposite has been reported [11]. Similarly to the reported effects of impending musculoskeletal injuries, between-race longitudinal factors produce more gradual adaptations over time; for example, resulting in small increases in stride frequency with training for two-year-old Thoroughbreds [12].

Within-race factors, such as changes in speed or onset of fatigue, affect stride parameters over the course of a race. Increases in speed are associated with increases in both stride length and stride frequency [13], while fatigue, typically observed towards the end of a race, leads to decreases in speed, stride length and stride frequency [14,15]. As racetrack curvature has been shown to affect speed in Thoroughbred racehorses [3], it was hypothesized that racetrack curvature also influences stride parameters within a race.

Most research into horse gallop stride parameters has been focused on Thoroughbred racehorses ridden by a jockey [8,10,11,12,13,14,15]. These horses need to support the weight of the jockey against gravity [16], and perform work to accelerate the jockey’s body mass in addition to their own [17]. In contrast, during Chuckwagon racing, the focus of our current study, four unmounted Thoroughbred horses, attached to a Chuckwagon in a 2-by-2 formation pull a heavy wagon around a racetrack at high-speed gallop. In this sport, a team of four horses, called an outfit, is harnessed in a paired formation, where the front pair are called leaders, and the back pair are called wheelers. The wheelers are attached to the wagon with tugs from the collar to the tree. The pole runs between the wheelers and is attached to their collars by the neck yoke which has two trees where leaders’ tugs are attached. Chuckwagon racing is a popular racing discipline mainly carried out in the Canadian provinces of Alberta and Saskatchewan over an intense period between May and August annually. The timed races start with a figure-of-eight manoeuvre around two barrels in the infield area, used for rodeo activities during non-race periods, followed by a race around the typically 5-furlong track. While Chuckwagon horses do not need to support any external vertical loads, they instead together must pull a 600 kg Chuckwagon [18]. Because of the different types of loading on the horse it is possible that fundamental associations between speed, curve exercise and stride parameters are different for ‘unmounted’ Thoroughbred Chuckwagon horses and Thoroughbred horses ridden by a jockey, similar to the increased stride frequency and decreased stance time that has been observed in Standardbreds in trot while pulling a load [19].

The present study aimed to investigate the association between racetrack curvature, speed, and stride parameters (stride frequency and stride length) for ‘unmounted’, wagon-pulling Thoroughbred Chuckwagon horses. It was hypothesized that, in alignment with previous studies in humans during sprint exercise [4,5,6], during curve running, speed and stride length would be lower, and stride frequency would be unaffected compared to straight-line running at equivalent speeds.

## 2. Materials and Methods

### 2.1. Ethical Approval

The study protocol was approved by the University of Calgary Animal Care and Use Committee (Approval number, AC23-0010).

### 2.2. Data Collection and Horses

Data was collected during three days of Chuckwagon training at the High River Ag Society 5/8th mile horseracing dirt racetrack in High River, Alberta, Canada between 30 April and 8 May 2025. Nine Chuckwagon outfits each consisting of four Thoroughbred horses were equipped with GNSS (Global Navigation Satellite System) loggers. On day one, two 10 Hz GNSS loggers (VBOX Sport, Racelogic, Novi, MI, USA) were used per horse, one placed over the withers area attached with insulation tape to the collar and one in the cranial/mid-thoracic area attached with insulation tape to the back pad. Four Chuckwagon outfits were instrumented on day one. On day two and three, a total of five outfits were equipped with 25 Hz GNSS loggers (RaceBox Mini S, RaceBox Motorsport LLC, Altamonte Springs, FL, USA) attached to the back pad with insulation tape. Equipment malfunction (battery failure) on day one necessitated a switch from the 10 Hz loggers to the 25 Hz data loggers on day two and three.

The total data set contains data from 28 Thoroughbred Chuckwagon horses (aged 4–15 years, all geldings, 8 left leaders (left horse in front row of 2-by-2 formation), 8 right leaders (right horse in front row), 7 left wheelers (left horse in back row), 5 right wheelers (right horse in back row)). For horses that had multiple measurements taken across the three days, only the first measurement is included in the analysis. Of the 28 horses in the data set, six were measured using collar-mounted GNSS loggers (all on day one), and 22 using back pad-mounted GNSS loggers.

### 2.3. Experimental Protocol

GNSS loggers were attached to the harness of the horses (left and right leader, left and right wheeler) before the harnesses were attached to the horses. On each day, loggers remained attached throughout consecutive data collection sessions until the last outfit had finished their training session. For each training session, data collection was initiated prior to the horses being hooked up to the wagon. This was achieved either by pressing the record button on the device (VBOX) or via a Bluetooth connection from an iOS device (iPhone 14 Pro, Apple Inc. Cupertino, CA, USA) positioned in the vicinity of the horse via the associated app (RaceBox). Data collection was stopped when the outfit returned from the training session (VBOX: button; RaceBox: Bluetooth connection via app).

A brief warmup of variable duration was conducted on the racetrack as deemed appropriate by each driver, typically consisting of medium speed trot and/or canter exercise of variable length. After the warmup period, a first trial run was conducted consisting of a figure of eight exercise around two barrels positioned in the “infield” of the 5/8th mile track mimicking the situation encountered in a Chuckwagon race followed by an acceleration to moderate speed and a period of canter/gallop exercise on track before returning to the “infield” area. This was followed by a second trial again navigating in a figure of eight around the “infield” barrels, this time followed by vigorous acceleration to near racing speed and typically one full lap around the half mile track. Each horse participated in a maximum of one training session per day.

### 2.4. Data Processing

At the end of each day, data were downloaded from the loggers. For the VBOX data loggers, data download was achieved from the SD card. For the RaceBox loggers, data were first uploaded to the cloud via the app and then downloaded to a laptop computer via the web interface. 10 Hz GNSS data (VBOX) included UTC time of day (in s) with one decimal place, latitude and longitude (in decimal minutes) with six decimal places, and velocity (in km/h) with two decimal places. Data exported from the 25 Hz GNSS loggers (RaceBox) included UTC time (date and time in hours, minutes and seconds) with two decimal places, latitude and longitude (in decimal degrees) with seven decimal places, and velocity (in m/s) with two decimal places.

Data analysis was carried out in MATLAB (version R2025b, The Mathworks Inc., Natick, MA, USA) according to the following steps:Speed and time conversion: speed values from VBOX were divided by 3.6 to convert from km/h into m/s. Date–time values from RaceBox were converted to POSIX time in seconds using built-in MATLAB function “*convertTo*”. For both VBOX and RaceBox data, time values were then converted to time elapsed since the start of the measurement, by subtracting the time at which the measurement was started.Extract gallop data: sections of data were identified, where all horses in an outfit exceeded speeds of 10 m/s, effectively restricting data analysis to portions of the data corresponding to gallop exercise excluding the initial ‘figure of eight’ exercise conducted in the ‘infield’ area.Correct for measurement errors: occasionally zero-speed values were recorded during a gallop section, sometimes followed by a compensatory speed increase. When a zero-speed value was identified, this value and the five subsequent speed values were set to NaN (Not a Number). The resulting gap in speed data was filled using autoregressive modelling from the built-in MATLAB function “*fillgaps*”.Accumulative distance travelled: the distance between consecutive GNSS datapoints was calculated based on the Haversine formula [20] as
(1)$$d=2R\sin^{-1}\left(\sqrt{\sin^{2}\left(\frac{\phi_{i+1}-\phi_{i}}{2}\right)+\cos\left(\phi_{i}\right)\cos\left(\phi_{i+1}\right)\sin^{2}\left(\frac{\lambda_{i+1}-\lambda_{i}}{2}\right)}\right)$$
where d is the distance between point i and i+1 in metres, $\phi_{i}$ and $\phi_{i+1}$ are the latitude values of point i and i+1 in radians, $\lambda_{i}$ and $\lambda_{i+1}$ are the longitude values of point i and i+1 in radians, and R is the radius of the Earth (equal to 6371 km [21]). The total distance covered since the start of the measurement was then found by cumulatively adding up the distances between all GNSS points.Change in heading: the heading $\theta_{i}$ in degrees at each GNSS point i was calculated based on the latitude and longitude of points i−1 and i+1, using MATLAB function “*azimuth*”. This function defines North as a heading of 0° and uses clockwise as positive direction. The heading change $\Delta \theta_{i}=\theta_{i}-\theta_{i-1}$ for each point i was then calculated as the difference in heading between consecutive GNSS points.Feature extraction per 100 m gallop section: the distance the horses covered was split up in sections of 100 m with an overlap of 80 m between sections effectively shifting a ‘100 m window’ [22] across the track with 20 m progression between consecutive windows. The following aggregate metrics were calculated per 100 m section:Average speed in m/s.Average stride frequency in Hz. The calculation of stride frequency followed the Fast Fourier Transform (FFT)-based approach described by Pfau et al. [22], which identifies the ‘dominant’ frequency based on a 6.4 s window (64 samples for the 10 Hz-logger, 160 samples for the 25 Hz-logger). The dominant frequency was identified based on maximum signal power, restricting the search to frequency bands between 1.5 and 3 Hz (restricting the search to values around the expected gallop stride frequency).Average stride length in m. Based on stride frequency (SF) and speed, stride length (SL) can be calculated [22]
(2)$$SL=\frac{speed}{SF}$$Absolute value of the curvature of the section in degrees. Calculated by applying a two-second averaging filter to the heading change Δθ and then summing it over the 100 m-section. According to the definition of the utilized MATLAB *azimuth*-function, clockwise turns have a positive curvature value and anticlockwise turns have a negative curvature value. Since all horses in this study navigated the track in anticlockwise direction, only absolute values of curvature are used.

Figure 1 provides an overview of an example data set extracted from an individual horse’s GNSS data logger as a means of illustrating the implemented data processing described above. In the left panel of Figure 1, each 100 m-section, with 20 m progression between subsequent sections, has been colour coded onto a track map showing a typical progression from the infield area around the track in an anticlockwise manner. The remaining panels illustrate speed (in m/s sampled at 25 Hz), as well as 100 m section average values for curvature (in degrees), stride length (in m) and stride frequency (in Hz) as time series. The series are colour coded according to the sections identified on the map. From these plots it becomes obvious that curvature values are increasing rapidly in the transition from a straight line to curved portions reaching maximum values of around 60 degrees for the example horse and rapidly dropping back towards zero when the horse is galloping on the straight. Fluctuations in stride length and frequency can be observed over the time series in a complex manner as a function of speed and curvature.

### 2.5. Statistics

To assess the effect of speed and track curvature on stride parameters, linear mixed models (SPSS v29, IBM, Armonk, NY, USA) were created with stride frequency and stride length (average values over 100 m sections) as output variables. The mixed models used continuous covariates speed (average over 100 m sections), curvature, and the two-way interaction between curvature and speed as fixed factors, and horse as random factor. Additionally, a linear mixed model was implemented to assess the influence of curvature on speed. This model used speed (average over 100 m-sections) as output variable, curvature as fixed factor, and horse as random factor. Findings were considered significant when *p* < 0.05.

## 3. Results

In the data set N = 1295 sections of 100 m were identified, each with an associated average speed, average stride length, and average stride frequency, see Figure 2. Median (first and third quartile) speed, stride length and stride frequency were 14.5 (12.9, 15.4) m/s, 6.32 (6.00, 6.58) m, and 2.265 (2.153, 2.361) Hz, respectively. Figure 2 illustrates the distribution of speed (Figure 2a), stride length (Figure 2b) and stride frequency (Figure 2c) across the 100 m-sections extracted from all horses’ GNSS data. Data points are colour coded from the first (black) to the last (white) 100 m section utilizing the MATLAB ‘hot’ colour coding. Refer to Figure 1 for the illustration of an example data set collected from one horse. The box plots demonstrate lower speed, stride length and stride frequency values towards the end of each training session (yellow data points) and the presence of data points with higher speed, stride length and stride frequency at the beginning (dark red) and in the middle (bright red and orange) of the training sessions. Curvature values calculated across all 100 m sections ranged between 0° and 64.7°, where low and high values are associated with straight and curved sections of track, respectively.

### 3.1. Effect of Track Curvature on Speed

Table 1 shows the effect of track curvature on speed, estimated by the linear mixed model. Curvature was found to be positively associated with speed (*p* = 0.004), meaning that on average speed increased with 0.0044 m/s for each degree increase in curvature or by about 0.26 m/s for a curvature value of 60 degrees typically observed during bend running in our study (see Figure 1 for an example).

### 3.2. Effect of Speed and Track Curvature on Stride Length

Table 2 shows the effect of speed, curvature, and the two-way interaction between curvature and speed on stride length, estimated by linear mixed model analysis. Speed (*p* < 0.001), curvature (*p* < 0.001), and the two-way interaction between curvature and speed (*p* < 0.001) all have a significant effect on stride length. Figure 3 shows a visualization of the linear mixed model for stride length as a function of curvature plotting stride length against curvature with speed values colour coded for each 100 m section. In addition to the data points, model lines of best fit are given for speed values equivalent to the first, second and third quartile (of 12.9, 14.5 and 15.4 m/s) of speed based on the coefficients provided in Table 2.

### 3.3. Effect of Speed and Track Curvature on Stride Frequency

Table 3 shows the effect of speed, curvature, and the two-way interaction between curvature and speed on stride frequency, estimated by linear mixed model analysis. Speed (*p* < 0.001), curvature (*p* < 0.001), and the two-way interaction between curvature and speed (*p* < 0.001) all have a significant effect on stride length. Figure 4 shows a visualization of the linear mixed model for stride frequency as a function of curvature plotting stride frequency against curvature with speed values colour coded for each 100 m section. In addition to the data points, model lines of best fit are given for speed values equivalent to the first, second and third quartile (of 12.9, 14.5 and 15.4 m/s) of speed based on the coefficients provided in Table 3.

## 4. Discussion

This study investigated the association between racetrack curvature and stride parameters in Thoroughbred Chuckwagon horses. The results showed that both stride length and stride frequency are statistically significantly associated with curvature. It appears important to provide some context for interpreting the small, statistically significant effects reported in our current study allowing the reader to judge the biological relevance of such small effects. At the median speed of 14.5 m/s and a curvature of 60° per 100 m section (a typical curvature value observed throughout the curve at the High River Ag Society horseracing track used in the present study, Figure 1), the linear mixed models predict an increase in gallop stride frequency of 0.053 Hz (+2.4%) from a value of 2.246 Hz on the straight to a value of 2.299 Hz during bend running. Stride length is predicted to change from 6.445 m on the straight to 6.308 m during bend running, a decrease by 0.137 m (−2.1%). This latter value is in the same order of magnitude as the previously reported 0.10 m decrease in stride length over consecutive races associated with an increase in injury risk by 11% [8]. While the previously published study was limited to investigating stride parameters on the final straight, our data suggests it might be beneficial to account for racetrack curvature when using stride parameter data from full races in injury prediction models. It should be noted, however, that the previous study was conducted on turf and synthetic racetracks in Thoroughbred flat racing [8], while our study used a dirt racetrack prepared for a Chuckwagon training session. Consequently, the values presented in our current study can not be applied to different racing disciplines or racing conditions.

Contrary to the findings by Tan and Wilson that maximum speed for Thoroughbred horses decreases during curve running [3], our results showed an increase in speed in the curve compared to the straight. At a curvature of 60° per 100 m section, the linear mixed model predicts an increase in speed of 0.264 m/s (Table 1) compared to a straight section of track. A possible explanation for this unexpected result might be the fact that our data was collected during training exercise, and consequently the horses were not running at their maximum speed. Additionally, fatigue, causing the horses to slow down considerably towards the end of each run, is likely a confounding factor. We have illustrated the distribution of speed, stride length and stride frequency for a typical training exercise (Figure 1) as well as across all data points collected from the 28 horses in the current study clearly illustrating the reduction in speed, stride length and stride frequency for the 100 m sections at the end of the training exercise (Figure 2). As the horses started each run with a curved section and ended with a straight section (Figure 1), this might be an explanation for the higher speeds observed in the curve. It may also be postulated that the addition of a jockey, adding vertical loads but showing an out-of-phase movement in the horizontal plane [17], might lead to differences in speed-curvature effects between unridden horses pulling a Chuckwagon and jockey-ridden horses. Finally, the type of surface might play a role in this context, with our study using a dirt surface prepared for Chuckwagon racing and the previous study using a turf surface for Polo or Thoroughbred racing [3]. Future research should clarify the biological relevance of the very small increase in speed reported here for curved track sections; for example, by conducting investigations into associations between speed, curvature, and stride parameters under Chuckwagon racing conditions.

The line plots in Figure 3 and Figure 4 illustrate trends in stride parameters for different speeds. As expected, both stride length and stride frequency increase with increasing speed. Stride length decreases in the curve compared to the straight, but the amount is dependent on the speed: at a speed of 12.9 m/s, stride length decreases with 1.04 mm per degree of curvature, while at 15.4 m/s it decreases with 2.99 mm per degree of curvature, i.e., more than twice the decrease at the higher speed. Similarly, stride frequency increases in the curve with 5.2 × 10^−4^ Hz per degree of curvature at 12.9 m/s, while at 15.4 m/s it approximately doubles to a value of 1.1 × 10^−3^ Hz per degree of curvature.

As the linear mixed models calculate stride parameter changes per degree of track curvature, they can potentially be applied to different racetracks with varying curve radii. Due to this generalizability, we propose to use these linear mixed models as part of future discipline-specific injury prediction models. However, the current model only contains data from one specific dirt racetrack prepared for Chuckwagon racing, and more generalizable Chuckwagon racing specific models should investigate a number of different tracks and add quantitative surface characteristics to the data set.

### Study Limitations

During Thoroughbred flat racing, typically GNSS devices are affixed to the saddle pad in the area behind the saddle [10,11]. The four horses pulling a Chuckwagon are attached to the wagon by means of a harness. These can be of varying design, but typically feature a collar fitted around the neck and a ‘back pad’ or ‘back strap’ fitted across the dorsal thoracic area of the horse. On the first day of data collection, we evaluated the practicality of attachment to the collar and to the back pad, subjectively judging the possibility of interference with the reins through conversation with the Chuckwagon drivers recruited to the study. From day two onward, GNSS loggers were only attached to the back pad. RaceBox Mini S devices were used instead of the VBOX Sport loggers throughout all experiments from day two and three, as the absence of buttons on these devices avoided unintentional shutting off of the loggers. The potential for interference (of the reins) with the button, combined with battery issues on the first day and associated data loss, has led to some inconsistency in the origin of valid GNSS data sets on day one, with some originating from collar-mounted and some from back pad-mounted loggers. For five of the horses, measurements were available from both the collar- and back pad-mounted loggers. When comparing stride parameters derived from both GNSS loggers, mean absolute differences were found to be 0.9% for speed, 0.4% for stride frequency, 0.9% for stride length and 1.7° for curvature. Future and ongoing studies are now exclusively utilizing the back pad (or back strap) mounting. Our study relies on the use of ‘consumer-grade’ GNSS devices with a sampling frequency of between 10 and 25 Hz. We have previously found a precision value of 0.0091 Hz for calculating stride frequency from a Fourier transformation-based approach over time periods similar to the ones used here [22]. Combined with the approach of averaging speed and stride parameter calculations over 100 m sections we are confident that the general trends identified in our current study are robust estimates.

Another limitation of the study is that data from horses from only two Chuckwagon drivers were used in this study, albeit with a number of different horses utilized for each driver. However, to better understand possible influences of differences in Chuckwagon or harness designs on GNSS derived stride parameters, it would be beneficial to collect data from a bigger sample size of Chuckwagon outfits.

Lastly, racetrack surface properties have been shown to significantly affect stride parameters [10], but they were not taken into account in this study. Some surface measurements were obtained during data collection that can be utilized as a reference for future data collection efforts on different Chuckwagon racetracks.

## 5. Conclusions

Our study reports that during gallop exercise, Thoroughbred Chuckwagon horses have a significantly shorter stride length and higher stride frequency in curved sections of the racetrack compared to the straight. The magnitude of these changes depends on the curvature of the racetrack and the speed of the horse. During the training sessions investigated here, speed in the curve was significantly higher than speed on the straight. However, this effect is likely confounded by fatigue, as each run started with a curved section and ended on a straight. Future studies should investigate this speed-curvature relation under racing conditions. At the median speed of 14.5 m/s and a curvature of 60° per 100 m section, a stride frequency increase of 0.053 Hz (+2.4%) and a stride length reduction of 0.137 m (−2.1%) was found compared to a straight section of track. This decrease in stride length is in the same order of magnitude as the previously reported stride length decrease of 0.10 m over consecutive races associated with an increase in injury risk by 11% [8]. This further highlights the relevance of detailed stride parameter models for prospective injury prediction. The linear mixed models presented in this study are a first step towards creating a discipline-specific stride parameter model incorporating racetrack geometry.

## Figures and Tables

**Figure 1 sensors-25-07376-f001:**
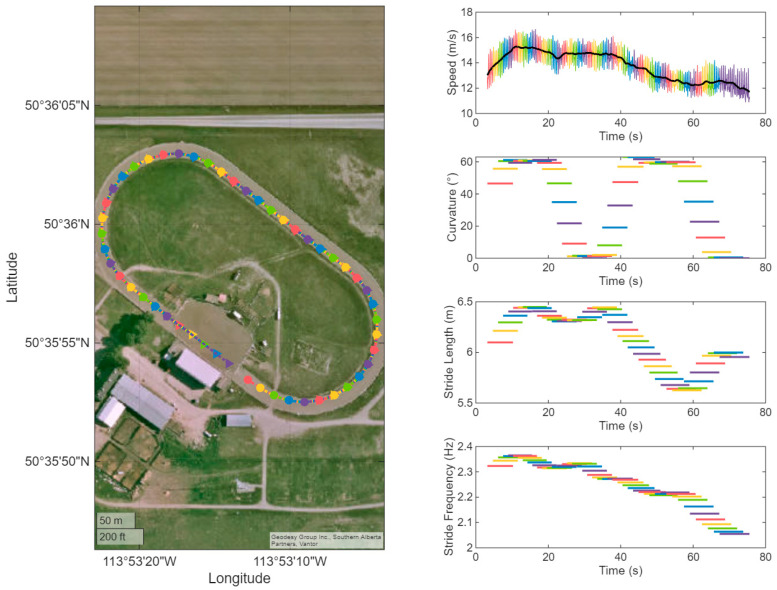
Example data calculated from a GNSS data logger attached to a Chuckwagon horse during training. Provided is an overview of the track with indication of latitude and longitude (left panel) as well as individual panels with speed (top right panel; in m/s), curvature (second panel from top right; in degrees), stride length (second panel from bottom right; in m) and stride frequency (bottom right panel; in Hz). Speed is shown at the original sample rate of 25 Hz; the remaining values are the average values for each of the 100 m sections colour coded from the first to the last section shown in the left panel. Sections have a length of 100 m, and run from a circular marker until the next marker with the same colour in the left panel.

**Figure 2 sensors-25-07376-f002:**
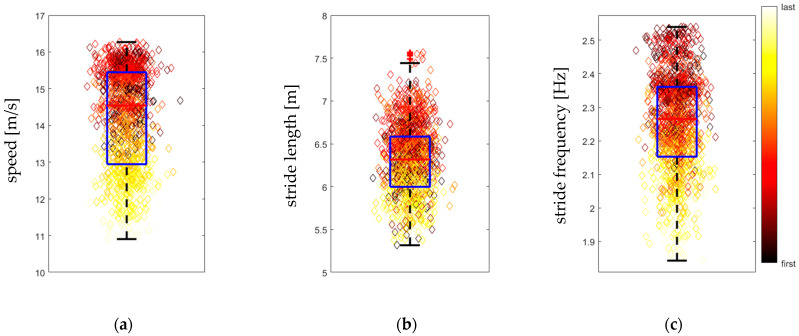
Boxplots of average speed (**a**), average stride length (**b**) and average stride frequency (**c**) for N = 1295 sections of 100 m length, recorded in 28 Thoroughbred Chuckwagon horses. Boxes represent first, second (red line) and third quartile. Whiskers extend to the furthest datapoint not considered an outlier. Outliers are datapoints further than 1.5 times the interquartile range away from the edge of the box and are indicated by a red plus-sign. Individual data points are colour coded according to the provided scale (MATLAB ‘hot’) from the first to the last 100 m section from which corresponding data points have been generated. Horizontal position of data points is varied according to a normal distribution in order to avoid overlapping data points.

**Figure 3 sensors-25-07376-f003:**
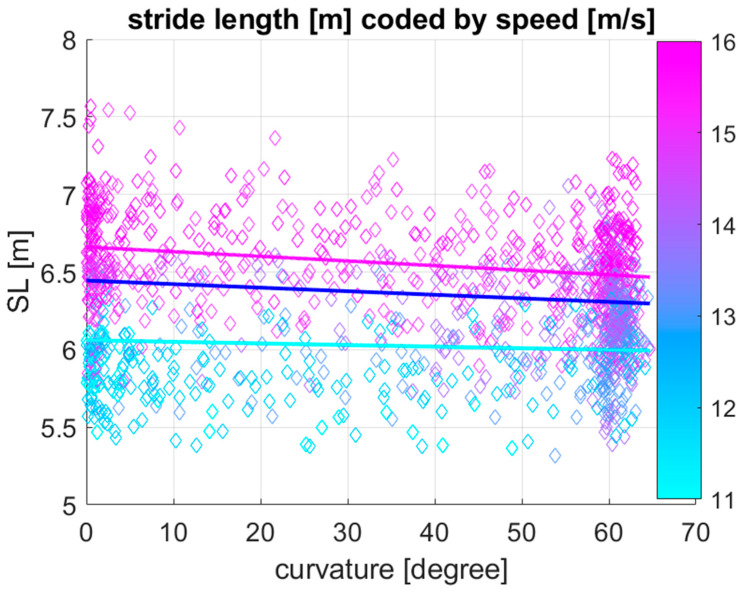
Visualization of the *N* = 1295 data points together with the outcome of the linear mixed model, predicting stride length (SL) as a function of curvature at three different speeds (Q1: 12.9 m/s, Q2: 14.5 m/s and Q3: 15.4 m/s of the speed boxplot in Figure 2a) for 28 Thoroughbred Chuckwagon horses. Stride length increases with increasing speed (colour coded from cyan to magenta), and it decreases with increasing curvature along the *x*-axis from near-zero curvature values on the straight to curvature values around 60 degrees for 100 m sections along curved portions of the track. Associated with the significant two-way interaction between curvature and speed, the stride length decreases more rapidly with increasing curvature for higher speed values.

**Figure 4 sensors-25-07376-f004:**
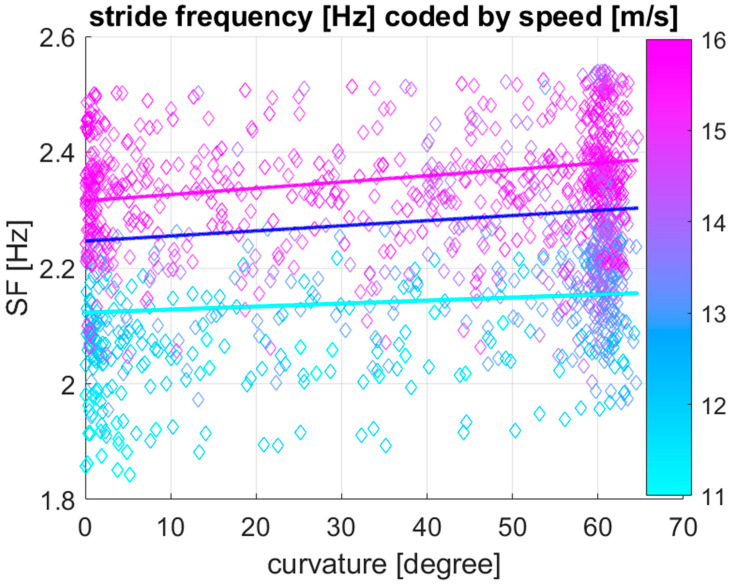
Visualization of the *N* = 1295 data points together with the outcome of the linear mixed model, predicting stride frequency (SF) as a function of curvature at three different speeds (Q1: 12.9 m/s, Q2: 14.5 m/s and Q3: 15.4 m/s of the speed boxplot in Figure 2a) for 28 Thoroughbred Chuckwagon horses. Stride frequency increases with increasing speed (colour coded from cyan to magenta), and it increases with increasing curvature along the *x*-axis from near-zero curvature values on the straight to curvature values around 60 degrees for 100 m sections along curved portions of the track. Associated with the significant two-way interaction between curvature and speed, the stride frequency increases more rapidly with increasing curvature for higher speed values.

**Table 1 sensors-25-07376-t001:** Estimates and 95%-confidence intervals of the fixed effect curvature on speed.

	*p*-Value	Fixed Effect Estimate	95% CI
Curvature	0.004	+0.0044 m/s per degree	(+0.0014, +0.0073) m/s per degree
Intercept	<0.001	14.07 m/s	(13.86, 14.27) m/s

**Table 2 sensors-25-07376-t002:** Estimates and 95% confidence intervals of the fixed effects speed and curvature on stride length.

	*p*-Value	Fixed Effect Estimate	95% CI
Speed	<0.001	+0.240 m per m/s	(0.234, 0.247) m per m/s
Curvature	<0.001	+9.02 mm per degree	(6.44, 11.6) mm per degree
Curvature × speed	<0.001	−0.78 mm per (degree × m/s)	(−0.96, −0.60) mm per (degree × m/s)
Intercept	<0.001	2.965 m	(2.833, 3.097) m

**Table 3 sensors-25-07376-t003:** Estimates and 95%-confidence intervals of the fixed effects speed and curvature on stride frequency.

	*p*-Value	Fixed Effect Estimate	95% CI
Speed	<0.001	+0.0772 Hz per m/s	(0.0748, 0.0796) Hz per m/s
Curvature	<0.001	−2.41 × 10^−3^ Hz per degree	(−3.33 × 10^−3^, −1.49 × 10^−3^) Hz per degree
Curvature × speed	<0.001	+2.27 × 10^−4^ Hz per (degree × m/s)	(1.63 × 10^−4^, 2.91 × 10^−4^) Hz per (degree × m/s)
Intercept	<0.001	1.127 Hz	(1.080, 1.174) Hz

## Data Availability

The data file providing speed, stride frequency, stride length and track curvature for 100 m sections of the track for all horses can be downloaded at: https://doi.org/10.6084/m9.figshare.30556235.

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
