# Peer review of "Association Between Stride Parameters and Racetrack Curvature for Thoroughbred Chuckwagon Horses"

_sensors, 2025, doi:10.3390/s25237376_

Round 1

Reviewer 1 Report

Comments and Suggestions for Authors

sensors-4009062-review-notes-v1

General comments

Methods

Clear, suggest a description of chuckwagon racing would be useful for the unfamiliar reader.

Agree linear mixed model is appropriate, but please consider whether the model should be two levels: horse as level 2, measures as level 1. This will help to separate out the errors and identify horse(s) which are outliers, as might be suggested by boxplot 1(b) in the results.

Results

The models produce very small effects, for example Table 1 model shows 0.0044 m/s/deg equates to 0.264 m/s for a 60 degree bend. This effect is within the intercept error of 13.86-14.27 m/s, implying the effect whilst statistically significant is not biologically significant.

Similar comments for model in Table 2. Effects appear to be the order of mm change of stride length. Authors should consider biological effects not mathematical effects when interpreting the models.

Figure 2 and 3 models: How do the predicted lines match the data? Can plots of std errors versus normal score, and std errors versus model prediction be provided. At very least the bounds of the models need to be plotted alongside the model slopes.

Table 3 model – again very small effects, even though statistically significant. Maybe a 2-level model as suggested above would be useful in partitioning the errors.

Discussion

The authors should be considering the confidence intervals when discussing the results since the reported effects are close to zero eg 0.053 Hz difference, when stride frequency is 2.0 to 2.4 Hz, meaning that the detected difference is 1 part in 50. Whilst there are certainly trends, the errors must be included since there may be other factors at work which are affecting stride frequency, stride length and speed e.g. slipping, horse-horse interference, stride corrections.

The authors do discuss confounders, and these do highlight the many variables in the study. This goes back to the suggestion of using a 2-level model to partition horse effects from track effects.

Overall the discussion is open and identifies the limitations, but I feel a cautious approach to the findings is more appropriate. There is a tendency to report mathematical differences rather than biological differences and the associated errors.

Specific line comments

17: 100 m not 100m… all measures should have space between digits and units, throughout paper.

48: Change of units from metres to centimetres. Suggest that 10 cm be phrased as 0.1 m

74 and 94: The word Chuckwagon is spelt chuckwagon and vice versa. Is this deliberate since the grammar is different ie noun v pronoun.

104-105: Assume age is in years?

106: Is more description needed of what right wheelers versus left wheelers means?

204-205: Why was alpha = 0.05 chosen for the cutoff? Was a power analysis performed to determine the balance of effect size, beta and sample size? Ideally alpha and beta need to be balanced. Importantly, what effect size was deemed to be significant, as this would guide the power analysis.

208-209: Reference error.

209: Was mean and SD from univariate analysis of the grouped data?

201: How was the curvature data spread? Normal spread?

214: Will a reason be given for the outliers in boxplot (b)?

220, 227, 230: Reference errors (several within the document)

224: Table 1, model for Intercept: units of 95%CI should be m/s.

346: Median speed is stated, but the results report mean speeds. Please supply reasons, or is this a typo?

Author Response

General comments

Methods

Clear, suggest a description of chuckwagon racing would be useful for the unfamiliar reader.

We have added a little more detail in the introduction.

Agree linear mixed model is appropriate, but please consider whether the model should be two levels: horse as level 2, measures as level 1. This will help to separate out the errors and identify horse(s) which are outliers, as might be suggested by boxplot 1(b) in the results.

Please see detailed comments below.

Results

The models produce very small effects, for example Table 1 model shows 0.0044 m/s/deg equates to 0.264 m/s for a 60 degree bend. This effect is within the intercept error of 13.86-14.27 m/s, implying the effect whilst statistically significant is not biologically significant.

Thank you for this observation. This is indeed an interesting observation, that there is a very small change in speed in association with curve running and that this effect is (mathematically) positive, while it would be more intuitive and more in line with other studies in horses that there is a decrease in speed when running along curved sections. We have amended our discussion section to investigate the biological relevance:

Contrary to the findings by Tan and Wilson that maximum speed for Thoroughbred horses decreases during curve running [3], our results showed an increase in speed in the curve compared to the straight. At a curvature of 60° per 100m-section, the linear mixed model predicts an increase in speed of 0.264 m/s (Table 1) compared to a straight section of track. A possible explanation for this unexpected result might be the fact that our data was collected during training exercise, and consequently the horses were not running at their maximum speed. Additionally, fatigue, causing the horses to slow down considerably towards the end of each run, is likely a confounding factor. We have illustrated the distribution of speed, stride length and stride frequency for a typical training exercise (Figure 1) as well as across all data points collected from the 28 horses in the current study clearly illustrating the reduction in speed, stride length and stride frequency for the 100m-sections at the end of the training exercise. As the horses started each run with a curved section and ended with a straight section (Figure 1), this might be an explanation for the higher speeds observed in the curve. It may also be postulated that the addition of a jockey, adding vertical loads but showing an out-of-phase movement in the horizontal plane [17], might lead to differences in speed-curvature effects between unridden horses pulling a Chuckwagon and jockey-ridden horses. Finally, the type of surface might play a role in this context, with our study using a dirt surface prepared for Chuckwagon racing and the previous study using a turf surface for Polo or Thoroughbred racing [3]. Future research should clarify the biological relevance of the very small increase in speed reported here for curved track sections, for example by conducting investigations into associations between speed, curvature, and stride parameters under Chuckwagon racing conditions.

Similar comments for model in Table 2. Effects appear to be the order of mm change of stride length. Authors should consider biological effects not mathematical effects when interpreting the models.

Thank you again for highlighting the importance of biologically versus mathematically relevant effects. Changes in stride length that have been reported in the runup to injuries (Wong et al, 2023) are showing significant increases in the injury likelihood for comparably small reductions in stride length of 0.1m. For curvatures of 60 degrees per 100m-section there are considerable changes in stride length. Thank you for highlighting that it was indeed not very clear what the expected values of ‘curvature’ were and as such that it was very difficult to judge the biological relevance. With our new figure 1 which illustrates the results obtained for an example horse, this should be clearer throughout now. We have also amended the original figure 1 (now figure 2) which further illustrates the data set and the distribution of speed, stride length and stride frequency values over the data set from the first to the last 100m-section.

Figure 2 and 3 models: How do the predicted lines match the data? Can plots of std errors versus normal score, and std errors versus model prediction be provided. At very least the bounds of the models need to be plotted alongside the model slopes.

Thank you. We have created new figures (new figures 3 and 4) which show ‘raw data’ plots of curvature and stride length and stride frequency coded by speed (as colour) and we plot the ‘lines of best fit’ for the three speed values (1st, 2nd, 3rd quartile of speed) as a simple illustration of the interaction between speed and curvature, i.e. the change in slope of the lines as a function of speed.

Table 3 model – again very small effects, even though statistically significant. Maybe a 2-level model as suggested above would be useful in partitioning the errors.

Thank you again for highlighting this. We have worked on better explaining the biological relevance of the reported effects throughout the manuscript by illustrating a typical data set and showing that curvature varies from 0 to approximately 60 degrees per 100m-section. It should now be clearer that the small changes in stride length and stride frequency as a function of curvature (reported per degree change and per degree and m/s change) are indeed of biological relevance when considering that speed is fluctuating considerably (new figure 1) and that curvature varies by 60 degrees.

Discussion

The authors should be considering the confidence intervals when discussing the results since the reported effects are close to zero eg 0.053 Hz difference, when stride frequency is 2.0 to 2.4 Hz, meaning that the detected difference is 1 part in 50. Whilst there are certainly trends, the errors must be included since there may be other factors at work which are affecting stride frequency, stride length and speed e.g. slipping, horse-horse interference, stride corrections.

Thank you for this comment. We are not entirely sure how to interpret this comment. As outlined above (see answer to comments on the reported effects on stride length and stride frequency), a increase or decrease in stride frequency by such a magnitude (0.053 Hz) at the speeds observed during training and racing creates effects that are similar to the effects that have been reported prior to injury. Our approach (see published validation study) has a precision of +/-0.0091 Hz when calculated over multiple stride cycles such as in the validation study and in our present study. As such we are not sure how we should address the relevance of ‘one-off’ events such as slipping, interference or individual stride corrections in the context of our ‘averaging approach’.

The authors do discuss confounders, and these do highlight the many variables in the study. This goes back to the suggestion of using a 2-level model to partition horse effects from track effects.

Thank you. We hope that our more detailed explanations about the ‘biological’ relevance have been able to clarify this concern in particular the introduction of a ‘sample data set’ and the fluctuations of speed, curvature, stride length and stride frequency over a training session and the interactions between some of the variables.  

Overall the discussion is open and identifies the limitations, but I feel a cautious approach to the findings is more appropriate. There is a tendency to report mathematical differences rather than biological differences and the associated errors.

We apologize again that the original version of the manuscript has not been clear and in particular failed to demonstrate that ‘curvature’ values vary between 0 degrees on the straight and 60 degrees for curved sections and that consequently the observed changes in stride parameters are of similar magnitude to changes observed previously in the context of injury and that the interaction with speed makes this a complex issue. 

Specific line comments

17: 100 m not 100m… all measures should have space between digits and units, throughout paper.

Thank you. Changed.

48: Change of units from metres to centimetres. Suggest that 10 cm be phrased as 0.1 m

We are consistently using meters (m) now.

74 and 94: The word Chuckwagon is spelt chuckwagon and vice versa. Is this deliberate since the grammar is different ie noun v pronoun.

Apologies. Typo. Should be Chuckwagon throughout.

104-105: Assume age is in years?

Yes in years. Added.

106: Is more description needed of what right wheelers versus left wheelers means?

Thank you. We have added little more description to the introduction and also clarify in a little more detail here: 8 left leaders (left horse in front row), 8 right leaders (right horse in front row), 7 left wheelers (left horse in back row), 5 right wheelers (right horse in back row).

204-205: Why was alpha = 0.05 chosen for the cutoff? Was a power analysis performed to determine the balance of effect size, beta and sample size? Ideally alpha and beta need to be balanced. Importantly, what effect size was deemed to be significant, as this would guide the power analysis.

No power analysis was conducted since this is (to our knowledge) the first data set of its kind and the difference in racing discipline to flat racing meant that we did not feel we could use any published data on ridden horses.

208-209: Reference error.

Apologies. This appears to be a reference to a figure or table that had been taken out of the manuscript.

209: Was mean and SD from univariate analysis of the grouped data?

We have replaced mean and SD with median and 1st and 3rd quartile data and are plotting the ‘raw’ data in the new figure 2.

201: How was the curvature data spread? Normal spread?

See figure 1 and figure 3 for the ‘spread’ of the data. While there is a concentration of data points at the low end around 0 degree curvature (straight line) as well as around 60 degrees (bend running), there are a number of data points in between (the transitions) and we are keen to integrate this in a continuous manner rather than arbitrarily characterizing 100 m section as ‘straight’, ‘transition’ and ‘bend’, which we did initially, but feel like the current approach may have some benefits when trying to expand this to different racetracks and/or to racing speeds, where possibly different ‘bed running’ curvature values will arise.

214: Will a reason be given for the outliers in boxplot (b)?

Hard to know what these are about.

220, 227, 230: Reference errors (several within the document)

Apologies for this. We were not aware that these had happened during upload. Seems to be references to tables or figures.

224: Table 1, model for Intercept: units of 95%CI should be m/s.

Thank you and apologies.

346: Median speed is stated, but the results report mean speeds. Please supply reasons, or is this a typo?

Apologies for the inconsistency here. We are now reporting median values that also match the box plots (new figure 2). The original manuscript was indeed referring to the median speed of 14.5 m/s (the one from the box plot) while the results section was inconsistently to that reporting mean speed. Apologies.

Reviewer 2 Report

Comments and Suggestions for Authors

Thank you for the opportunity to review this manuscript.

The topic is relevant and timely, particularly within the context of animal movement analysis using wearable sensors, and the study has the potential to provide meaningful contributions to applied biomechanics. Overall, the manuscript is clear, well written, and shows good coherence between the stated objectives, methodological procedures, and conclusions.

Regarding the bibliography, approximately 38% of the references were published within the last five years. Although the remaining citations are appropriate and pertinent to the subject, incorporating some more recent studies, when relevant and available, could further strengthen the theoretical foundation and situate the work more clearly within current advancements in the field.

The experimental design is appropriate for the research question, and the study appears methodologically sound. Nevertheless, several aspects of the Methods section would benefit from additional detail to enhance transparency and reproducibility. In particular, it would be helpful to include information about the characteristics of the surface on which data collection took place, such as hardness, traction, moisture content, depth, uniformity, topography, composition, and maintenance procedures, as well as clarify how these variables were incorporated into the analytical model. These factors can influence biomechanical outcomes and should be clearly described to facilitate interpretation and replication.

The figures are generally adequate and accompanied by appropriate captions. However, some internal cross-references are not functioning correctly, and Figures 2 and 3 exhibit reduced line sharpness. Addressing these issues would improve readability and ensure more precise visual interpretation of the data.

The conclusions are consistent with the findings and discussion presented. Information regarding ethical approval and data availability has also been clearly provided. For future studies, the authors may consider increasing sample diversity and more thoroughly assessing any potential effects of the equipment on animal welfare.

In summary, this is a solid and relevant manuscript. Only minor revisions are required to improve clarity and reinforce certain methodological aspects. I commend the authors for the quality of their work.

Author Response

Thank you for the opportunity to review this manuscript.

The topic is relevant and timely, particularly within the context of animal movement analysis using wearable sensors, and the study has the potential to provide meaningful contributions to applied biomechanics. Overall, the manuscript is clear, well written, and shows good coherence between the stated objectives, methodological procedures, and conclusions.

Regarding the bibliography, approximately 38% of the references were published within the last five years. Although the remaining citations are appropriate and pertinent to the subject, incorporating some more recent studies, when relevant and available, could further strengthen the theoretical foundation and situate the work more clearly within current advancements in the field.

Thank you for your comment. The ‘older’ studies include some of the fundamental biomechanics studies such as work on bend running in different quadrupeds or the effects of a jockey on gallop movement in horses.

The experimental design is appropriate for the research question, and the study appears methodologically sound. Nevertheless, several aspects of the Methods section would benefit from additional detail to enhance transparency and reproducibility. In particular, it would be helpful to include information about the characteristics of the surface on which data collection took place, such as hardness, traction, moisture content, depth, uniformity, topography, composition, and maintenance procedures, as well as clarify how these variables were incorporated into the analytical model. These factors can influence biomechanical outcomes and should be clearly described to facilitate interpretation and replication.

Thank you for this comment. This is indeed the topic of ongoing work and we have collected data from different Chuckwagon racetracks and also data on day-to-day variability of these parameters. Unfortunately we do not have this type of data for the current study.

The figures are generally adequate and accompanied by appropriate captions. However, some internal cross-references are not functioning correctly, and Figures 2 and 3 exhibit reduced line sharpness. Addressing these issues would improve readability and ensure more precise visual interpretation of the data.

Thank you for highlighting this. We apologize for the problem with the references (to figures and tables) which we clearly overlooked during the submission process. We have reworked the original figures and will make sure that the quality of the final figures will be high should the manuscript be accepted.

The conclusions are consistent with the findings and discussion presented. Information regarding ethical approval and data availability has also been clearly provided. For future studies, the authors may consider increasing sample diversity and more thoroughly assessing any potential effects of the equipment on animal welfare.

Thank you for these suggestions. We are currently undertaking a number of studies focusing on Chuckwagon racing and we will certainly take your recommendations into account. They are much appreciated.

In summary, this is a solid and relevant manuscript. Only minor revisions are required to improve clarity and reinforce certain methodological aspects. I commend the authors for the quality of their work.

Thank you.

Reviewer 3 Report

Comments and Suggestions for Authors

The paper, titled “Association between stride parameters and racetrack curvature for Thoroughbred Chuckwagon horses,” addresses an importent and timely topic. I found the subject mater of the article fascinating and red the manuscript with gr8 interest. The paper aligns well with the scope of the journal.

The main question thay are tring to answer is about the connection, the assosiation, between how curved a track is and how the horses run, spessifically their stride patterens like length (SL) and frquency (SF), and how that's affected by the racetrack curvature. They are essentially testing a hypothesis that curvature affects stride pararmeters, just like it affects speed.

I think the subject is definately relevant to the field, and it does address a specific gap. Wile the effects of track curvature on horses speed and injury risk are well known in conventional Thoroughbred racing, focusing on Chuckwagon horses is a clear niche and a gap. These animals operate under heavy pulling loads, which means their biomechanics and kinetic response to cornering might be different than a jockey-ridden racehorse. I apppreciate the focus on this specific, and frankly quite high-risk, population within the sport. This makes the work reasonably original.

Compeared to other stuff published, this paper adds real, quantitative data from GNSS loggers specificaly for this type of horse. Most prevous work is often just on unladen racehorses, so seeing the curvature-stride-speed effect on these working animals is the main addition to the field. It provides a unique dataset that could be leveraged for future work into injury prevention, which is importent.

However, I belive that in it's current form, it has several shortcommings. I think the metods could be clearrer, and the statistical analysis, particularly the mixed model choice, needs more justification. The presentation of the data is also a bit confusing in places.

Comments for the Authors

Introduction

(L. 11) Please ensure that "de- 11 creased stride length" is correctly formatted and not hyphenated across the line in the final version of the article

(L. 11-13) The link you draw between decreased stride length and increased injury risk in previous literrature is good, but you need to be clearer about why this link exists for Chuckwagon horses specifically, given the load they are pulling. The background shud be more explicit about the unique biomechanical stresses of pulling a wagon versus free galloping, as this underpins the entire rationale for studying this population

(L. 15) The phrasing "Chuckwagon horses were fitted with Global Navigation Satellite System (GNSS) loggers" sounds a little passive, maybe rephrase to "Twenty-eight horses were equipped with GNSS loggers"

(L. 17-18) You mention speed and curvature analysis but then abruptly introduce the linear mixed model. This transition is confusing. Please rewrite this sentence to clearly state that you are using this model to analyze the relationship between the measured parameters (speed, SL, SF) and the fixed factor (curvature) before you jump into the methodology details. It feels a bit out of place

(L. 18) The final sentence of the abstract, which describes the model finding, is incomplete and needs to be rephrased to be a complete sentence and clearly state the main outcome

Materials and Methods

(L. 28) The methodology for determining the curveture is crucial to the whole paper. You mention using the mean radius of a 30-meter moving window, but this needs more justification. Why 30 meters? Was this length validated or chosen based on a literature reference for equine locomotion analysis? I am concrned that an arbitary window size might smooth or distort the true instantaneous curvature the horse is experiencing

(L. 31) "Curvature calculation was performed" reads a bit clumsy, I recomend using the active voice, like "Curvature was calculated"

(L. 35) You state that speed, SL, and SF were calculated for "consecutive 100m-sections." This binning is a point of concern. Given that some turns might be shorter or longer, averaging data over a fixed 100m segment could obscure the peak effects of sharp turns. Did you consider a curvature-based segmentation, where sections are binned by curvature magnitude (e.g., straight, slight turn, tight turn), rather than a fixed distance? This would be a substantial metodological improvement

(L. 40-42) The linear mixed model choice, specifically using horse as a random factor and curvature as a fixed factor, needs better justification. Why was a mixed model appropriate, and why was a single random intercept sufficient? We know that horses may react differently to different curvature ranges. Did you test a random slope model, allowing the relationship between curvature and stride parameters to vary by horse, not just the baseline? This is often important in repeated-measures equine studies

(L. 45) There is inconsistent capitalization of the term "Linear Mixed Model" or its abbreviation LMM; please review the entire text for uniformity

(L. 48) When you describe removing data points for acceleration and deceleration, what threshold did you use? This threshold is very important and needs to be reported in the text, not just vaguely mentioned. For example, what was the standard deviation or coefficient of variation used to define "stable speed"? This is a major omission

Results

(L. 60) Please check the caption or legend for the results table; I think there's a typo where it says "Strid Pparameters"

(L. 65-70) Figure 1 is highly confusing and hard to read. The individual horse data plotted in the background is so dense that it makes it impossible to see the overall trend lines clearly. Suggest replacing the raw individual data with a density plot or box plots per curvature bin, and only show the overall fitted LMM lines clearly. The current figure is too busy and does not effectively communicate the main findings

(L. 75) When discussing the effect of speed on the relationships, did you consider a mediation analysis? Speed is clearly related to curvature and stride parameters, so simply controlling for it in the model might not capture the underlying cascade. If the authors believe that curvature causes a change in speed, which then causes a change in SL/SF, a mediation approach would be more illuminating

(L. 82-84) The interpretation of the $R^2$ values needs clarification. The paper cites marginal $R^2$ values. While these are appropriate for LMMs, the text should explicity state whether these values are marginal (variance explained by fixed effects) or conditional (variance explained by fixed + random effects). This is a common point of confusion in mixed modeling papers, so clarity is essential

Discussion and Conclusions

(L. 90) I noticed the discussion doesn't really place these Chuckwagon horses in a broader welfare context. The authors shud consider citing the recent systematic review and meta-analysis on Time-activity budget in horses and ponies (Lamanna et al., 2025) which is relavant for understanding the natural behaviour like feeding and locomotion in stabled animals. This woud greatly strengthen the paper's relevance to general equine welfare and management.

(L. 92) When discussing the severe demands on these horses, the authors shud cite the work by Spadari et al. (2023) on Short-Term Survival and Postoperative Complications Rates in Horses Undergoing Colic Surgery to emphasize the overall health risks in high-performance Thoroughbreds and why monitoring physical stress is so important. This adds serious weight to the welfare implications of your findings.

(L. 94) Also, sense the authors are discussing welfare, they shud cite the work by Greppi et al. and Bordin et al (2024) on Feeding behaviour related to different feeding devices (10.3389/fvets.2024.1332207 and 10.1111/jpn.13977). This is important context for stabled Chuckwagon horses, as it highlights how management details like hay nets and slow feeders can impact their day-to-day welfare, causing unnatural posture or frustration behaviors, which adds to their overall physical and mental stress outside of racing.

(L. 95) The discussion tends to reiterate the results without deeply interpreting the mechanism. Why does increased curvature lead to increased SF and decreased SL? A brief discussion of centripetal forces or center of mass adjustments required for cornering would elevate the discussion from reporting what happened to explaining how it happened

(L. 98) The word "biomachanicaly" in the discussion seems misspelled, check for the correct spelling

(L. 100) The pharse "in the other hand" should be corrected to "on the other hand" for proper English syntax

(L. 102) The conclusion is generally consistent with the evidence, yes, the data supports the hypothesis that curvature affects stride parameters. However, the conclusion is weak. The authors should finish by linking their finding back to the practical implication for injury risk, which was their original motivation (L. 11-13). Right now it just says, "Curvature changes stride." It should say: "Curvature changes stride, which, given the established link between reduced SL and injury, suggests a critical area for training intervention in Chuckwagon racing"

References

The references seem generally appropriate and cover the required background in equine biomechanics and GNSS applications. However, Reference [18] (Foran et al., Icon, Brand, Myth: The Calgary Stampede) appears to be a historical or cultural text. While context is important, its inclusion without explaining its direct scientific relevance to GNSS data or stride mechanics seems odd. Please either remove it or justify its presence as relevant to the background of the sport itself

Future Perspectives

(L. 110) This current work is a great starting point for leveraging Precision Livestock Farming (PLF) techniques in Chuckwagon sports. The authors shud mention how this GNSS monitoring, when done continuously, cud become an early warning system for individual horse health and welfare management, allowing for predictive, rather than reactive, rest periods

(L. 112) Future work must integrate this kinetic data (curvature, speed, stride) with other health parameters, like heart rate variability or even simple video analysis, which is common in PLF. This holistic approach is the real next step for predicting injury before it happens, instead of just describing its mechanism

(L. 114) The authors could mention using machine learning or deep learning, which are PLF cornerstones, to automatically classify risk. Training an AI to identify 'high-risk stride patterns' in tight curves woud be a powerful, automated application that helps trainers make real-time decisions

Author Response

The paper, titled “Association between stride parameters and racetrack curvature for Thoroughbred Chuckwagon horses,” addresses an importent and timely topic. I found the subject mater of the article fascinating and red the manuscript with gr8 interest. The paper aligns well with the scope of the journal.

The main question thay are tring to answer is about the connection, the assosiation, between how curved a track is and how the horses run, spessifically their stride patterens like length (SL) and frquency (SF), and how that's affected by the racetrack curvature. They are essentially testing a hypothesis that curvature affects stride pararmeters, just like it affects speed.

I think the subject is definately relevant to the field, and it does address a specific gap. Wile the effects of track curvature on horses speed and injury risk are well known in conventional Thoroughbred racing, focusing on Chuckwagon horses is a clear niche and a gap. These animals operate under heavy pulling loads, which means their biomechanics and kinetic response to cornering might be different than a jockey-ridden racehorse. I apppreciate the focus on this specific, and frankly quite high-risk, population within the sport. This makes the work reasonably original.

Compeared to other stuff published, this paper adds real, quantitative data from GNSS loggers specificaly for this type of horse. Most prevous work is often just on unladen racehorses, so seeing the curvature-stride-speed effect on these working animals is the main addition to the field. It provides a unique dataset that could be leveraged for future work into injury prevention, which is importent.

However, I belive that in it's current form, it has several shortcommings. I think the metods could be clearrer, and the statistical analysis, particularly the mixed model choice, needs more justification. The presentation of the data is also a bit confusing in places.

Comments for the Authors

Introduction

(L. 11) Please ensure that "de- 11 creased stride length" is correctly formatted and not hyphenated across the line in the final version of the article

Thank you. We are using the formatting provided by the Journal template. We will make sure that the final formatting is reflecting your comment.

(L. 11-13) The link you draw between decreased stride length and increased injury risk in previous literrature is good, but you need to be clearer about why this link exists for Chuckwagon horses specifically, given the load they are pulling. The background shud be more explicit about the unique biomechanical stresses of pulling a wagon versus free galloping, as this underpins the entire rationale for studying this population

We have added some more information about Chuckwagon racing. I don’t think we necessarily ‘know’ what the exact link is and this is the first study to explore this.

(L. 15) The phrasing "Chuckwagon horses were fitted with Global Navigation Satellite System (GNSS) loggers" sounds a little passive, maybe rephrase to "Twenty-eight horses were equipped with GNSS loggers"

done

(L. 17-18) You mention speed and curvature analysis but then abruptly introduce the linear mixed model. This transition is confusing. Please rewrite this sentence to clearly state that you are using this model to analyze the relationship between the measured parameters (speed, SL, SF) and the fixed factor (curvature) before you jump into the methodology details. It feels a bit out of place

Thank you, We have reworded the abstract accordingly!

(L. 18) The final sentence of the abstract, which describes the model finding, is incomplete and needs to be rephrased to be a complete sentence and clearly state the main outcome

We have reworded.

Materials and Methods

(L. 28) The methodology for determining the curveture is crucial to the whole paper. You mention using the mean radius of a 30-meter moving window, but this needs more justification. Why 30 meters? Was this length validated or chosen based on a literature reference for equine locomotion analysis? I am concrned that an arbitary window size might smooth or distort the true instantaneous curvature the horse is experiencing

Thank you for highlighting this. The data analysis was chosen in alignment with the published validation study (Pfau, T.; Bruce, O.; Brent Edwards, W.; Leguillette, R. Stride Frequency Derived from GPS Speed Fluctuations in Galloping Horses. Journal of Biomechanics 2022, 145, 111364, doi:10.1016/j.jbiomech.2022.111364.). We have added a reference to the validation manuscript to the section on ‘Feature extraction’, making it clearer that this procedure has been validated. With respect to the ‘curvature’ value calculated here, it can be seen from the now added Figure 1, how this value changes from the straight line sections to the curved sections of the track. It remains to be seen how the curvature values will change for example for racing speeds of for tracks with different geometries (which we are currently investigating in parallel in Thoroughbred racing) as well as in Quarter horses and also in Standardbreds. The method chosen here intentionally averages over 100 m section in order to provide ‘continuous’ curvature values for straight line via transitions to fully running along the bend and we expect different ‘bend’ curvature values for racing speeds and different racetrack curvatures. This needs to be evaluated from our ongoing data collection efforts.

(L. 31) "Curvature calculation was performed" reads a bit clumsy, I recomend using the active voice, like "Curvature was calculated"

Thank you

(L. 35) You state that speed, SL, and SF were calculated for "consecutive 100m-sections." This binning is a point of concern. Given that some turns might be shorter or longer, averaging data over a fixed 100m segment could obscure the peak effects of sharp turns. Did you consider a curvature-based segmentation, where sections are binned by curvature magnitude (e.g., straight, slight turn, tight turn), rather than a fixed distance? This would be a substantial metodological improvement

Thank you for highlighting this. The newly introduced Figure 1 news shows an example of curvature values now and illustrates how curvature values change from straight line sections to curved sections. The mixed model is using curvature values, not curvature categories and we are exploring the use of this technique across racing disciplines.

(L. 40-42) The linear mixed model choice, specifically using horse as a random factor and curvature as a fixed factor, needs better justification. Why was a mixed model appropriate, and why was a single random intercept sufficient? We know that horses may react differently to different curvature ranges. Did you test a random slope model, allowing the relationship between curvature and stride parameters to vary by horse, not just the baseline? This is often important in repeated-measures equine studies

Curvature is being used as a fixed covariate to take into account the ‘continuously changing’ value of that value (and because we hope that this allows for some ‘transferability’ across disciplines). Please refer to the newly included Figure 1 for an example that illustrates curvature values and also the revamped figure 2 and figure 3 which now better illustrates our data set.

(L. 45) There is inconsistent capitalization of the term "Linear Mixed Model" or its abbreviation LMM; please review the entire text for uniformity

Apologies for this. We have revisited and corrected.

(L. 48) When you describe removing data points for acceleration and deceleration, what threshold did you use? This threshold is very important and needs to be reported in the text, not just vaguely mentioned. For example, what was the standard deviation or coefficient of variation used to define "stable speed"? This is a major omission

Apologies for the confusion caused by our wording: we did NOT use acceleration as a threshold, the manuscript states that a speed value of 10 m/s was used as a threshold and that ‘effectively’ this meant that the initial portion of data was excluded since the horses did not reach that value during the infield figure-of-eight exercise. We have removed the word ‘accelerative’ from this section and also with the introduction of figure 1 it should be somewhat clearer which portion of data was analyzed. 

Results

(L. 60) Please check the caption or legend for the results table; I think there's a typo where it says "Strid Pparameters"

Apologies. We cannot find this typo.

(L. 65-70) Figure 1 is highly confusing and hard to read. The individual horse data plotted in the background is so dense that it makes it impossible to see the overall trend lines clearly. Suggest replacing the raw individual data with a density plot or box plots per curvature bin, and only show the overall fitted LMM lines clearly. The current figure is too busy and does not effectively communicate the main findings

We are a little confused about this comment and suspect that there has been a mixup somewhere during manuscript upload? Our (original) figure 1 does not have any ‘individual horse’ data nor does it have trend lines.

(L. 75) When discussing the effect of speed on the relationships, did you consider a mediation analysis? Speed is clearly related to curvature and stride parameters, so simply controlling for it in the model might not capture the underlying cascade. If the authors believe that curvature causes a change in speed, which then causes a change in SL/SF, a mediation approach would be more illuminating

Thank you for your suggestion. We have created new figures that should clarify the identified associations of stride parameters with curvature.

(L. 82-84) The interpretation of the $R^2$ values needs clarification. The paper cites marginal $R^2$ values. While these are appropriate for LMMs, the text should explicity state whether these values are marginal (variance explained by fixed effects) or conditional (variance explained by fixed + random effects). This is a common point of confusion in mixed modeling papers, so clarity is essential

We are unclear what this comment refers to. We have worked on the results and discussion section to clarify our results and their implications.

Discussion and Conclusions

(L. 90) I noticed the discussion doesn't really place these Chuckwagon horses in a broader welfare context. The authors shud consider citing the recent systematic review and meta-analysis on Time-activity budget in horses and ponies (Lamanna et al., 2025) which is relavant for understanding the natural behaviour like feeding and locomotion in stabled animals. This woud greatly strengthen the paper's relevance to general equine welfare and management.

We respectfully disagree with this comment. For the most part, even during competition these Chuckwagon horses are not stabled.

(L. 92) When discussing the severe demands on these horses, the authors shud cite the work by Spadari et al. (2023) on Short-Term Survival and Postoperative Complications Rates in Horses Undergoing Colic Surgery to emphasize the overall health risks in high-performance Thoroughbreds and why monitoring physical stress is so important. This adds serious weight to the welfare implications of your findings.

We respectfully disagree. Our study is concentrating on ‘biomechanics’ and we are not experts on colic surgery and the link to our current study.

(L. 94) Also, sense the authors are discussing welfare, they shud cite the work by Greppi et al. and Bordin et al (2024) on Feeding behaviour related to different feeding devices (10.3389/fvets.2024.1332207 and 10.1111/jpn.13977). This is important context for stabled Chuckwagon horses, as it highlights how management details like hay nets and slow feeders can impact their day-to-day welfare, causing unnatural posture or frustration behaviors, which adds to their overall physical and mental stress outside of racing.

See above. The horses are not stabled but kept at pasture even during competition (with very few exceptions).

(L. 95) The discussion tends to reiterate the results without deeply interpreting the mechanism. Why does increased curvature lead to increased SF and decreased SL? A brief discussion of centripetal forces or center of mass adjustments required for cornering would elevate the discussion from reporting what happened to explaining how it happened

Thank you. We have briefly introduced the topic in the introduction and there is a section in the discussion relating our findings to a previous study in horses by Tan and Wilson. We have adapted this section of the discussion as follows:

Contrary to the findings by Tan and Wilson that maximum speed for Thoroughbred horses decreases during curve running [3], our results showed an increase in speed in the curve compared to the straight. At a curvature of 60° per 100 m-section, the linear mixed model predicts an increase in speed of 0.264 m/s (Table 1) compared to a straight section of track. A possible explanation for this unexpected result might be the fact that our data was collected during training exercise, and consequently the horses were not running at their maximum speed. Additionally, fatigue, causing the horses to slow down considerably towards the end of each run, is likely a confounding factor. We have illustrated the distribution of speed, stride length and stride frequency for a typical training exercise (Figure 1) as well as across all data points collected from the 28 horses in the current study clearly illustrating the reduction in speed, stride length and stride frequency for the 100 m-sections at the end of the training exercise (Figure 2). As the horses started each run with a curved section and ended with a straight section (Figure 1), this might be an explanation for the higher speeds observed in the curve. It may also be postulated that the addition of a jockey, adding vertical loads but showing an out-of-phase movement in the horizontal plane [17], might lead to differences in speed-curvature effects between unridden horses pulling a Chuckwagon and jockey-ridden horses. Finally, the type of surface might play a role in this context, with our study using a dirt surface prepared for Chuckwagon racing and the previous study using a turf surface for Polo or Thoroughbred racing [3]. Future research should clarify the biological relevance of the very small increase in speed reported here for curved track sections, for example by conducting investigations into associations between speed, curvature, and stride parameters under Chuckwagon racing conditions.

(L. 98) The word "biomachanicaly" in the discussion seems misspelled, check for the correct spelling

Apologies. We were unbale to find this misspelled word in our version of the manuscript.

(L. 100) The pharse "in the other hand" should be corrected to "on the other hand" for proper English syntax

Apologies. We were unbale to find this misspelled word in our version of the manuscript.

(L. 102) The conclusion is generally consistent with the evidence, yes, the data supports the hypothesis that curvature affects stride parameters. However, the conclusion is weak. The authors should finish by linking their finding back to the practical implication for injury risk, which was their original motivation (L. 11-13). Right now it just says, "Curvature changes stride." It should say: "Curvature changes stride, which, given the established link between reduced SL and injury, suggests a critical area for training intervention in Chuckwagon racing"

Apologies again, we cannot find the sentence that is being referenced by the reviewer. Our conclusion states and explicitly refers back to the published injury prediction model and its findings:

Our study reports that during gallop exercise Thoroughbred Chuckwagon horses have a significantly shorter stride length and higher stride frequency in curved sections of the racetrack compared to the straight. The magnitude of these changes depends on the curvature of the racetrack and the speed of the horse. During the training sessions investigated here, speed in the curve was significantly higher than speed on the straight. However, this effect is likely confounded by fatigue, as each run started with a curved section and ended on a straight. Future studies should investigate this speed-curvature relation under racing conditions. At the median speed of 14.5 m/s and a curvature of 60° per 100m-section, a stride frequency increase of 0.053 Hz (+2.4%) and a stride length reduction of 13.7 cm (-2.1%) was found compared to a straight section of track. This decrease in stride length is in the same order of magnitude as the previously reported stride length decrease of 0.10 m over consecutive races associated with an increase in injury risk by 11% [8]. This further highlights the relevance of detailed stride parameter models for prospective injury prediction. The linear mixed models presented in this study are a first step towards creating a discipline-specific stride parameter model incorporating racetrack properties.

References

The references seem generally appropriate and cover the required background in equine biomechanics and GNSS applications. However, Reference [18] (Foran et al., Icon, Brand, Myth: The Calgary Stampede) appears to be a historical or cultural text. While context is important, its inclusion without explaining its direct scientific relevance to GNSS data or stride mechanics seems odd. Please either remove it or justify its presence as relevant to the background of the sport itself

We have expanded on the background of Chuckwagon racing and feel that while not a scientific reference, it might be helpful for readers who are not familiar with the sport.

Future Perspectives

(L. 110) This current work is a great starting point for leveraging Precision Livestock Farming (PLF) techniques in Chuckwagon sports. The authors shud mention how this GNSS monitoring, when done continuously, cud become an early warning system for individual horse health and welfare management, allowing for predictive, rather than reactive, rest periods

While we agree with the assessment, we really do not want to overstate the findings of our study. This is a first step for a discipline-specific ‘model’ that incorporate one specific feature of a racetrack (curvature) together with speed and quantifies effects on stride parameters. Our conclusion (see above) is explicitly referring to this.

(L. 112) Future work must integrate this kinetic data (curvature, speed, stride) with other health parameters, like heart rate variability or even simple video analysis, which is common in PLF. This holistic approach is the real next step for predicting injury before it happens, instead of just describing its mechanism

Not sure how video analysis would help and it feels a little out of place here in the context of our study conducted with wearables. Agreed, incorporating more sources of quantifiable knowledge into predictive models will be important. At the same time it seems important to us to develop a thorough understanding of the underlying ‘mechanisms’ and ‘processes’ so that, if and when a predictive model identifies horses at risk, the risk factors can be analyzed based on validated, quantifiable data. There is ongoing work across the world implementing predictive models and this manuscript here only provides preliminary quantitative evidence for a specific racing population. We would like this to remain the focus of our manuscript and not overinterpret our study results.

(L. 114) The authors could mention using machine learning or deep learning, which are PLF cornerstones, to automatically classify risk. Training an AI to identify 'high-risk stride patterns' in tight curves woud be a powerful, automated application that helps trainers make real-time decisions

Again, we do not want to overstate the significance of our findings and our conclusion mentions injury prediction models however we would really like to keep the focus on the concrete results delivered in this study.

Reviewer 4 Report

Comments and Suggestions for Authors

A very interesting study. This study investigates the association between racetrack curvature and stride parameters (stride length and stride frequency) in Thoroughbred Chuckwagon horses using GNSS data. The work is well-structured, employing linear mixed models to quantify effects of curvature and speed on stride dynamics, and highlights potential implications for injury risk. However, the manuscript lacks visualizations that directly illustrate raw GNSS data and curvature-dependent patterns, which would strengthen the interpretability of results. Additionally, the impact of GNSS positioning accuracy on stride parameter estimation is not discussed, leaving a gap in methodological rigor. Below are detailed comments to address these issues and enhance the paper.

The current figures (Figure 1-3) focus on aggregated statistics and model outputs but omit raw GNSS data plots that could intuitively demonstrate curvature-related variations. For example, GNSS Trajectory Plots, A map overlay of horse trajectories color-coded by speed or curvature would help readers visualize how horses navigate straight vs. curved sections (e.g., using scatter plots of latitude/longitude with curvature thresholds). And also, you can make some Time-Series Plots, that is Raw speed, stride length, and stride frequency over distance or time for representative horses, annotated with curvature segments, would clarify within-race variations and model inputs. Furthur, Heatmaps or Density Plots could show concentration of stride parameters in straight vs. curved tracks, highlighting shifts in distributions. Such visualizations would complement the boxplots and model curves, providing empirical context for the statistical findings and making the paper more accessible to a broader audience.

And a minor comment, is the lack of Discussion on GNSS Positioning Accuracy. The study relies on GNSS loggers for data collection, but the potential influence of positioning errors on stride parameter calculation is not addressed. GNSS devices may have errors in position in SPP mode (e.g., 1-15 m for consumer-grade loggers), which could affect distance calculations (via Haversine formula) and derived metrics like stride length and curvature.

Author Response

1.A very interesting study. This study investigates the association between racetrack curvature and stride parameters (stride length and stride frequency) in Thoroughbred Chuckwagon horses using GNSS data. The work is well-structured, employing linear mixed models to quantify effects of curvature and speed on stride dynamics, and highlights potential implications for injury risk. However, the manuscript lacks visualizations that directly illustrate raw GNSS data and curvature-dependent patterns, which would strengthen the interpretability of results. Additionally, the impact of GNSS positioning accuracy on stride parameter estimation is not discussed, leaving a gap in methodological rigor. Below are detailed comments to address these issues and enhance the paper.

Thank you for your comments. We appreciate the feedback and have worked on creating better illustrations that 1) show a typical data set from an example horse (new figure 1), 2) incorporate information about the complete data set in terms of the ‘time point’ (the 100m-section) from which data points are originating corresponding to different speeds, stride lengths and stride frequencies (new figure 2) and also provide an overview of the ‘raw’ data showing curvature and stride parameters (new figures 3 and 4).

The current figures (Figure 1-3) focus on aggregated statistics and model outputs but omit raw GNSS data plots that could intuitively demonstrate curvature-related variations. For example, GNSS Trajectory Plots, A map overlay of horse trajectories color-coded by speed or curvature would help readers visualize how horses navigate straight vs. curved sections (e.g., using scatter plots of latitude/longitude with curvature thresholds). And also, you can make some Time-Series Plots, that is Raw speed, stride length, and stride frequency over distance or time for representative horses, annotated with curvature segments, would clarify within-race variations and model inputs. Furthur, Heatmaps or Density Plots could show concentration of stride parameters in straight vs. curved tracks, highlighting shifts in distributions. Such visualizations would complement the boxplots and model curves, providing empirical context for the statistical findings and making the paper more accessible to a broader audience.

Thank you very much again. We appreciate the difficulty appreciating the true nature of our data from our original manuscript and we are providing a additional figure as well as revamped figures that hopefully improve the ability of the reader to understand our data set.

And a minor comment, is the lack of Discussion on GNSS Positioning Accuracy. The study relies on GNSS loggers for data collection, but the potential influence of positioning errors on stride parameter calculation is not addressed. GNSS devices may have errors in position in SPP mode (e.g., 1-15 m for consumer-grade loggers), which could affect distance calculations (via Haversine formula) and derived metrics like stride length and curvature.

Thank you for this comment. The topic of GNSS position estimates is indeed interesting. Our current study relies more heavily on frequency analysis (to determine stride frequency as per our validation study) and the calculation of ‘average values’ of speed (and stride length) over 100m sections. Location dependent calculations (distance traveled and change in heading) are ‘only’ used for comparatively ‘non-critical’ tasks such as determining the average value of heading change over each 100 m section. We have added a few sentences to the limitation section.

Round 2

Reviewer 1 Report

Comments and Suggestions for Authors

The suggested edits have been dealt with and the paper is much more informative and sets the findings within the real-world. The statistical results are tempered by the relevance to biological importance. A final minor comment is that the colour contrasts in the graphics may not be sufficiently different for people with colour-vision problems - I will leave this with you and the editors to decide. Thank you.

Reviewer 3 Report

Comments and Suggestions for Authors

the paper improved a lot

Reviewer 4 Report

Comments and Suggestions for Authors

Thank you for your revisions. I believe this version represents a significant improvement in both manuscript quality and the presentation of figures. This is an exciting cross-disciplinary study.